# Crosstalk Between Sickle Cell Disease and Ferroptosis

**DOI:** 10.3390/ijms26083675

**Published:** 2025-04-13

**Authors:** Annamaria Russo, Giuseppe Tancredi Patanè, Antonella Calderaro, Davide Barreca, Ester Tellone, Stefano Putaggio

**Affiliations:** Department of Chemical, Biological, Pharmaceutical and Environmental Sciences, University of Messina, Viale Ferdinando Stagno d’Alcontres 31, 98166 Messina, Italy; arusso@unime.it (A.R.); anto.calderaro@gmail.com (A.C.); ester.tellone@unime.it (E.T.); stefano.putaggio@studenti.unime.it (S.P.)

**Keywords:** iron overload, cell death, oxidative stress, sickle cell hemoglobin, transfusion, system Xc^−^, GPx4

## Abstract

Sickle cell disease (SCD) is an inherited hemoglobin disorder that is widespread across the globe. It is characterized by a very complex pathogenesis, but at the basis of the disease is the mutation of the HBB gene, which determines the production of a mutated hemoglobin: sickle cell hemoglobin (HbS). The polymerization of HbS, which occurs when the protein is in a deoxygenated state, and the greater fragility of sickle cell red blood cells (sRBCs) determine the release of iron, free heme, and HbS in the blood, favoring oxidative stress and the production of reactive oxygen species (ROS). These features are common to the features of a new model of cell death known as ferroptosis, which is characterized by the increase of iron and ROS concentrations and by the inhibition of glutathione peroxidase 4 (GPx4) and the System Xc^−^. In this context, this review aims to discuss the potential molecular and biochemical pathways of ferroptosis involved in SCD, aiming to highlight possible tags involved in treating the disease and inhibiting ferroptosis.

## 1. Introduction

Sickle cell disease (SCD) is an inherited blood disorder prevalent worldwide, especially in regions of Africa, the Middle East, the Mediterranean, India, and Southeast Asia. It is related to the production of a mutated hemoglobin and involves an altered structure of red blood cells (RBCs), which take on a crescent or sickle shape [1,2,3]. Individuals with SCD experience a range of health problems, such as acute and chronic pain crises, an increased susceptibility to infection, a potential for organ damage, and a lower life expectancy. The remarkable prevalence of the disease is related to its genetic inheritance; therefore, genetic counseling is of extreme importance for the evaluation of disease transmission [4]. The main cause of the disease is the mutation of the gene responsible for the Hb mutation, the β-globin (HBB) gene, located on chromosome 11p15.5. The substitution of GTG nucleotide with GAG results in the replacement of hydrophilic glutamic acid (Glu) with hydrophobic valine (Val) and, thus, the production of a mutated hemoglobin, called sickle cell hemoglobin (HbS). HbS, unlike normal adult hemoglobin (HbA), tends to polymerize (at low oxygen concentrations), resulting in the deterioration of RBCs structure and increased fragility. The transit of erythrocytes within blood vessels causes cells to be exposed to oxygen-deficient tissue regions, resulting in the polymerization of HbS that alters erythrocyte structure (morphology) and increases cellular fragility [5,6,7]. Moreover, sickle red blood cells (sRBCs), unlike normal RBCs, have an extremely shortened average lifespan (about 20 days) due to increased hemolysis [8,9]. SCD, besides affecting erythrocytes, has damaging effects throughout the body, triggering several pathological conditions, including vessel occlusion, hemolytic anemia, vascular–endothelial dysfunction, inflammation, and more. [1]. These disease states lead to the activation of pro-oxidant enzymes, the release of Hb (which promotes the Fenton reaction), and the alteration of mitochondrial respiration, events that contribute to the increased production of free oxygen radicals (ROS) and, thus, oxidative stress [10]. Among the various therapeutic strategies that aim to alleviate the symptoms of the disease are transfusions. However, continuous transfusions can potentially result in an overload of free iron, contributing to increased ROS and lipid peroxides, processes also implicated in a mechanism of programmed cell death known as ferroptosis [11]. In this context, this review aims to investigate the potential contribution of ferroptosis in SCD. Ferroptosis is a form of cell death, first discovered by Stockwell and described by Dixon in 2012, which features iron accumulation, increased ROS, and lipid peroxidation as markers. This process is implicated in numerous disease states, such as neurodegenerative diseases, cardiovascular diseases, and more. [12]. Understanding the biochemical implications of ferroptosis in SCD progression could provide innovative insights for new therapies aimed at improving the health status of patients.

## 2. Pathophysiology of Sickle Cell Disease

The pathophysiology of SCD is generally related to the Hb mutation causing the sickling of erythrocytes, alteration of blood rheology, and tissue damage. In detail, the polymerization of HbS in a deoxygenated environment results in the tightening and increased density of the erythrocyte membrane, favoring its rupture; in addition, this condition can potentially lead to blood vessel occlusion, ischemia, infarction, and the hemolysis of RBCs [13]. Intravascular hemolysis causes increased Hb concentration at the blood level and accumulation of sodium and calcium ions within erythrocytes. This results in dysfunction of the calcium-dependent ATPase pump on the one hand and extracellular Hb increase on the other, reducing NO levels and promoting vasoconstriction [13,14,15]. The main mechanisms involved in the pathogenesis of SCD include HbS polymerization, vaso-occlusion, increased blood viscosity, endothelial dysfunction, ischemia reperfusion, oxidative stress, and inflammation (see Figure 1).

### 2.1. Polymerization of HbS and Deformation of sRBCs

As already mentioned, the primary event in the pathophysiology of SCD is the HBB genetic mutation resulting in the amino acid substitution, at position 6 in both β-chains, of Glu amino acid with Val. This substitution leads to the polymerization of deoxygenated HbS, which, unlike Hb, undergoes fiber formation under conditions of oxygen deprivation. Specifically, the deoxygenation to which sRBCs are exposed when they reach tissues with high oxygen demand results in a conformational change in HbS and the exposure of hydrophobic amino acid residues (Val). This condition leads to the interaction of the β-globin chains, which thus initiate the nucleation process, leading to the formation of the HbS polymer. The newly formed polymers possess a high growth rate and lead to the production of long fibrils of HbS, which contributes to the alteration of the erythrocyte component; in fact, they promote increased membrane stiffness, erythrocyte deformation, energetic insufficiency, stress conditions, and premature hemolysis [3,16,17]. It should be highlighted that the rate of HbS polymerization is affected by the concentration of HbS and fetal hemoglobin (HbF). HbF consists of 2 α chains and 2 γ chains (α-globin and γ-globin). High levels of HbF are implicated in reducing the concentration of HbS, slowing down the rate and tendency of polymerization; in addition, HbF cannot participate in the formation of the deoxygenated polymeric state of HbS. HbF can form hybrid structures with HbS, leading to the formation of tetramers (α2, βS, γ) that cannot form polymeric structures with HbS in the deoxygenated state. The anti-polymerizing action of HbF is related to two aminoacidic residues in the chain γ, glycine γ87 and aspartic acid γ80 [18]. A direct consequence of HbS polymerization is deformation of erythrocytes, which assume a sickle shape. The sRBCs are rigid, less deformable, and more prone to hemolysis than healthy red blood cells; they also cannot flow smoothly within blood vessels, causing vessel occlusion and tissue ischemia [19]. Furthermore, sickling of erythrocytes results in the exposure of phosphatidylserine, and facilitates the adhesion of sRBCs with other blood and endothelial components through basal cell adhesion molecule-1/Lutheran (B-CAM-1/Lu), integrin-associated protein (IAP), and intercellular adhesion molecule-4 (ICAM-4), adhesion molecules that are normally expressed in the extracellular region of the erythrocyte membrane [7]. Polymerization of HbS also results in increasing oxidative damage on the membrane and promoting ROS formation; in addition, cell lysis, triggered by both increased cell fragility due to HbS polymerization and oxidative damage results in the release of Hb into the bloodstream, which damages the vascular wall and activates markers of inflammation [20].

### 2.2. Occlusion of Vessels and Endothelial Dysfunction

Altered blood rheology, increased adhesiveness of RBCs with inflammatory cells and endothelial walls, and chronic hemolysis contribute to increased plasma viscosity and occlusion of blood vessels, causing painful systemic vaso-occlusive crisis (VOC) in individuals with SCD [21]. In addition, it should be noted that the altered morphology of sRBCs prevents the flow of cells within the smaller caliber blood vessels, such as capillaries and venules, due to their reduced ability to deform, thus preventing the flow of oxygen within tissues and leading to ischemia [17]. Intra-/extra-vascular hemolysis of sRBCs leads, in addition, to the increase in free Hb and arginase-1 at the vascular level, causing NO depletion, and triggering vasoconstriction, endothelial dysfunction, and vascular remodeling [22]; in addition, Hb and heme groups promote ROS production, erythrocyte and vessel membrane damage, promoting cell adhesion to the endothelium [1]. The disruption of sRBCs, endothelial dysfunction and inflammation can also potentially lead to endothelial activation and release of inflammatory factors, such as selectins (E- and P-), vascular cell adhesion molecules-1 (VCAM-1), ICAM-1, and key leukocyte chemoattractant, such as interleukin-8 (IL-8), which promote the release of vasoactive peptides, such as endothelin-1, that stimulate vasoconstriction [21,23,24]; in addition, the potential activation of neutrophils and platelets results in their increased affinity for the endothelial wall and sRBCs, contributing to vessel occlusion [25]. Individuals with SCD have elevated levels of neutrophils, monocytes and platelets, and circulating aggregates (neutrophils-platelets and/or monocytes and platelets) that correlate with disease severity [26]. Several studies showed the involvement of inflammatory stimuli in the initiation of vaso-occlusion in vivo in murine models [27,28,29,30]. It should also be pointed out that repeated episodes of vaso-occlusion and reperfusion during the restoration of normal blood flow can be involved in the manifestation of ischemia reperfusion, which leads to ROS generation and which stimulates inflammation and tissue damage, further implementing the consequences of SCD [11].

### 2.3. Oxidative Stress

Several studies show that the presence of free heme and iron, released by hemolysis of sRBCs, contributes to increased oxidative stress in subjects with SCD [7,11,31]. Increased ROS causes activation of the inflammatory process, endothelial dysfunction, and adhesion of sRBCs to vessel endothelium, further promoting vessel occlusion [32]. In SCD, ROS and reactive nitrogen species (RNS) are potentially generated by sRBCs, leukocytes, platelets, endothelial cells and plasma enzymes; moreover, several studies show that the production of these molecules is linked to different mechanisms such as increasing activity of nicotinamide-adenine dinucleotide phosphate (NADPH) oxidase and endothelial xanthine oxidase (XO), HbS auto-oxidation, heme and iron release, asymmetric dimethylarginine (ADMA), decoupling of nitric oxide synthase (NOS) activity and reduced bioavailability of NO [33,34,35,36,37,38].

#### 2.3.1. Oxidase Activity

As previously mentioned, oxidases are enzymes that can potentially lead to the production of ROS and RNS in individuals with SCD. Among these, NADPH oxidase, an enzyme involved in the generation of the superoxide anion ($O_{2}^{-}$) is the main oxidative enzyme of leukocytes, endothelial, and red blood cells [31]. The increase in ROS contributes to hemolysis associated with infections or occlusive crises of the vessels, to the activation of inflammatory and thrombogenic responses, and to the fragility of erythrocytes; in addition, in RBCs, the functionality of NADPH oxidase is modulated by intra- and extra-cellular factors, such as protein kinase C and Rac GTPases, or by extracellular signaling factors such as transforming growth factor β1 and endothelin-1 [39,40]. In addition, a study conducted by Aslan et al. showed how the XO enzyme, involved in the production of $O_{2}^{-}$ and hydrogen peroxide (H_2_O_2_), in conditions of hypoxia and reoxygenation, is released by the liver and reaches the circulation, altering vascular function and increasing the production of ROS [32].

#### 2.3.2. Autoxidation of HbS and Decrease in NO

Under normal conditions, during oxygenation/deoxygenation cycles, there is a slow autoxidation of Hb into methemoglobin (MetHb) (transition from the ferrous form to the ferric form) and the production of $O_{2}^{-}$ and H_2_O_2_ by erythrocytes. The reactive species produced during these processes are, however, neutralized by the enzymatic and non-enzymatic antioxidant systems present in the cells; in addition, methemoglobin reductase reduced back MetHb to Hb to maintain the stable oxidative balance in the cells [41]. However, in pathological conditions, early and chronic hemolysis of sRBCs results in the accumulation of HbS in the blood. HbS, compared with HbA, is more prone to autoxidation and unstable, especially under conditions of oxygen deprivation [42,43]. HbS polymerization is the main molecular event of hemolytic anemia in SCD also because of its potential pseudo-peroxidase activity; moreover, Kassa et al. showed the rate of self-reduction in HbA is higher than HbS, and that MetHb and free heme (product of oxidation) could be central in the development of SCD. These findings suggest the development of potential therapeutic treatments that aim to interrupt the inflammatory processes related to free heme. Researchers demonstrated that the free ferryl radical can induce the irreversible oxidation and dimerization of the cysteine residue 93 of the β-chain and damage the organelles of target cells, such as mitochondria, contributing to the increase in oxidative stress and favoring the progression of the disease [44]. Reiter et al. instead observed that MetHb, present in plasma in high concentrations in cases of SCD, can interact with NO, leading to the production of methemoglobin nitrate and iron-nitrosemoglobin complexes, which reduce the plasma concentration of NO, favoring vasoconstriction [44,45]. In cases of SCD the biological properties of NO, such as regulation of vascular tone and/or aggregation and adhesion in the vascular compartment, are impaired, due to the reduced bioavailability of the gas [46,47]; moreover, given the high presence of HbS, NO reacts with the protein leading to the formation of a very stable complex, Fe^2+^Hb-NO, which can potentially be involved in the Fenton reaction [48]. In addition to NO, the role of L-arginine should also be highlighted. Amino acid does not participate exclusively in the production of NO, but thanks to the action of arginase, it is directed toward the production of urea and ornithine. In cases of SCD, given the activation of the enzyme, the amino acid is subtracted from NOS (which uses it to produce NO), leading to a reduced bioavailability of the gas [49,50]. It should also be noted that in cases of SCD, the decoupling of NOS is favored due to the reduced concentration of L-arginine and tetrahydrobiopterin (BH4), due to the presence of peroxynitrite (ONOO^−^), a condition that further determines the deficiency of NO and the increase in oxidative stress and vaso-occlusion [51].

### 2.4. Iron

Under normal conditions, the Hb released by RBCs binds, in the blood, haptoglobin (Hp), leading to the formation of a highly stable complex that hinders the release of iron and promotes its elimination through internalization in macrophages, thanks to the presence of the CD163 receptor [52]. In pathological conditions, given the reduced viability of sRBCs, the blood concentration of HbS increases, hindering the normal activity of macrophages, which on the one hand store iron to eliminate it, and on the other release it at the level of the blood flow; moreover, given the hydrophobicity of the molecule, it tends to accumulate at the level of cell membranes, implementing the production of ROS. In detail, macrophages can have an inflammatory (M1) or anti-inflammatory (M2) phenotype; M1 is a toxic phenotype that can lead to the production of ROS and pro-inflammatory cytokines, while M2 is a protective phenotype that produces neurotrophic factors and anti-inflammatory cytokines [53]; in SCD, these cells exhibit an M1 phenotype, leading to increased iron uptake [54]. In addition, free heme, resulting from erythrocyte hemolysis, can react with $O_{2}^{-}$, H_2_O_2,_ and NO, producing ROS and RNS and stimulating the Fenton reaction (Fe^2+^ + H_2_O_2_ → Fe^3+^ + ∙OH + OH^−^) [55]. It should also be noted that iron overload in subjects with SCD is linked to the iron released by sRBCs, to chronic transfusions that cause an imbalance in iron homeostasis and potentially to an alteration of iron metabolism, conditions that lead to an imbalance of hepcidin [56]; in addition, studies in the literature show that individuals with SCD may experience iron overload due to chronic transfusions [57]. A clinical study conducted by van Beers et al. has highlighted how iron overload linked to transfusions causes serious diseases in different body districts, such as the heart, liver, and pancreas, inducing cytotoxicity on different cell lines [58]. It is therefore essential to monitor ferritin levels in individuals with SCD to reduce their concentrations through therapies that use iron chelators [59].

## 3. Ferroptosis

Ferroptosis is a new form of cell death, first discovered by Stockwell and described by Dixon in 2012, which differs from other cell deaths in both the biochemical pathways involved and the morphological features. Specifically, from a biochemical point of view, ferroptosis is characterized by altered iron homeostasis and Fe^2+^ accumulation, reduced GPx4 activity and GSH production, increased ROS generation, and lipid peroxidation; also, characteristics of the process are the morphological changes in the mitochondria, which undergo shrinkage, a reduction in the number of ridges, membrane thickening, and rupture of the outer mitochondrial membrane [12]. Specifically, GPx4 is a cell membrane-bound peroxidase and is involved in reducing the hydroperoxides of cell membrane phospholipids to their corresponding phospholipid alcohol. In RBCs, GPx4 regulates cellular redox homeostasis by protecting the cell from oxidative stress through the inhibition of lipid peroxidation and reduction in ROS [60]. In SCD, HbS polymerization increases RBCs’ fragility, driving premature cell death and lysis. The resulting Hb release causes an increase in free heme and iron that can be the onset of the ferroptotic process with vascular injury and cardiomyocyte death, promoting the development of several diseases.

### 3.1. Iron Metabolism and Mitochondrial Dysfunction

Iron is one of the major trace elements in the body and is involved in regulating numerous biological processes, such as DNA synthesis, mitochondrial respiration, and enzyme activity, etc. [61]. Iron, in its Fe^2+^ form (the main form found in the body), is transported into the circulation by transferrin (TF) binding and, through transferrin receptor 1 (TFR1), is internalized in cells and stored in endosomes [62,63,64]. Iron, in the endosomal environment, is reduced to Fe^3+^ by the metal reductase STEAP3 and subsequently released into the cytosol by member 2 of the solute transporter family 11 (SLC11A2), where it can potentially form labile iron complexes (LIPs), which are involved in triggering the ferroptotic process [65,66]. The increase in complexes triggers lipid peroxidation through the Fenton reaction and stimulates the activity of enzymes involved in ROS production, leading to ferroptosis [61]. Excess iron ions in the cytosol can on one hand be stored by ferritin, a cytosolic protein composed of two subunits (ferritin heavy chain 1 (FTH1) and ferritin light chain (FTL)), which is involved in iron elimination, and on the other hand can be eliminated through member 1 of the solute transporter family 40 (SLC40A1) [67]. Furthermore, it has been shown that ferritin degradation, due to activation of nuclear receptor 4 (NCOA4), can cause iron release and trigger ferroptosis [68]. It should also be emphasized that altering intracellular iron concentration also alters the function of mitochondria, organelles essential for cell survival; in fact, these represent an important site of iron utilization and can potentially be a major source of cellular ROS [69,70]. Iron imported into the mitochondria through the solute transporter family 25 member 37 (SLC25A37/mitoferrin-1) and the solute transporter family 25 member 28 (SLC25A28/mitoferrin-2) is used for heme biosynthesis and iron-sulfur (Fe-S) clusters [66]. Triggering ferroptosis results in an alteration of mitochondrial structure and energy metabolism. In fact, during ferroptosis, there is an increase in oxidative phosphorylation, ATP production and a decrease in glycolysis, mechanisms that lead to a cellular energy imbalance [71]. A study conducted by Oh-Jin-Oh et al. on different cellular lineages highlighted the role of voltage-dependent anion channels (VDAC) in the regulation of ferroptosis, showing that inhibition of transporter activity causes inhibition of the process and reduced production of mitochondrial ROS [72]. In addition, it has been observed how ferroptosis regulates the electron transport chain, inducing ROS production; in fact, the I-II-III complex of the mitochondrial respiratory chain (NADH-CoQ reductase complex, succinate dehydrogenase complex, cytochrome bc-1 complex, respectively) can produce ROS by damaging the mitochondrial membrane and triggering lipid peroxidation, inducing ferroptosis [64].

### 3.2. Biochemical Pathway Involved in Ferroptosis and Antioxidants System

The accumulation of cellular iron results, as mentioned earlier, in increased Fenton reaction, ROS accumulation, and increased lipid peroxidation [73]. Excess iron causes increased levels of non-transferrin-bound iron (NTBI), which through reaction with H_2_O_2_ and ferric ions leads to increased free radicals [74]. The ROS produced oxidized membrane polyunsaturated fatty acids (PUFAs) leading to the production of lipid peroxides (PUFAs-OOH) and the alteration of cell membrane stability; in addition, molecules resulting from the degradation of PUFAs-OOH, such as 4-hydroxynonenal (HNE) and malondialdehyde (MDA), can potentially form complexes with proteins and DNA altering their structure and functionality [75]. Enzymes involved in lipid peroxide synthesis include acyl-CoA synthetase, a member of the long-chain family 4 (ACSL4), and lysophosphatidylcholine acyltransferase 3 (LPCAT3) [76,77]. In detail, ACSL4 promotes PUFA-CoA synthesis by catalyzing a reaction between free PUFAs and acyl-CoA, and through the intervention of LPCAT3, the PUFA-CoAs form PL-PUFAs, molecules involved in cell membrane biosynthesis and potentially prone to peroxidation [77]. Another enzyme class involved in the peroxidation process is lipoxygenases (LOX), iron-containing enzymes that oxidize membrane PUFAs by positively modulating ferroptosis, leading to the production of radicals [77,78]. It should be emphasized that the body has a wide range of antioxidant agents, enzymatic and nonenzymatic in nature, which enable the neutralization of ROS. Among these, glutathione is certainly one of the most important endogenous molecules given its involvement in the biosynthesis of iron-sulfur clusters, its action as a substrate of transferases, and GPx [79]. In detail, GPxs include enzymes involved in the reduction in hydroperoxides, including lipid peroxides (LOOH) to H_2_O and derivative alcohols, by the oxidation of GSH to GSSG (disulfide form). There are 8 isoforms of the enzyme (GPx1 to GPx8), but the one involved in the ferroptotic process is represented by GPx4. The activity of the enzyme is modulated by GSH concentrations, which in turn depends on the activity of the System Xc^−^, a cystine/glutamate antiporter composed of two chains, the light chain SLC7A11 (Solute Carrier Family 7 Member 11) and the heavy chain SLC3A2 (Solute Carrier Family 3 Member 2). GSH synthesis occurs in the cytosol and involves glutamate, glycine, and cysteine (limiting factor for tripeptide synthesis). Under normal conditions, normal functions of the System Xc^−^ allow the cell to maintain a balance between glutamate and cysteine concentrations; inhibition of the System Xc^−^, on the other hand, brings a reduction in cystine entry into the cellular compartment, resulting in inhibition of GSH synthesis. This results in the reduction in GPx4 activity and alteration of the cell’s redox homeostasis, accumulation of ROS, and initiation of the ferroptotic process [80]. Recent studies show that there are compensatory pathways that allow cells to avoid going into ferroptosis, such as the pathway involving ferroptosis oxidoreductase suppressor protein 1 (FSP1); the enzyme reduces Coenzyme Q 10, inhibiting lipid peroxidation [81,82]. Another class of enzymes involved in the inhibition of ferroptosis is represented by PL lysophospholipid acyltransferase 1 (MBOAT1) and MBOAT2 [83]. Different studies show that GTP cyclohydrolase 1 inhibits ferroptosis by producing dihydrobiopterin and BH4; in detail, BH4 exerts a protective effect on PUFA tails by inhibiting oxidative degradation [84]. Nuclear factor erythroid 2-related factor 2 (Nrf2) is also involved in the inhibition of ferroptosis through the stimulation of ferroptosis target genes represented by FTH1, quinone oxidoreductase 1 (NQO1), and heme oxygenase-1 (HO-1) [85].

## 4. Correlation Between Ferroptosis and Sickle Cell Disease

To date, the etiopathogenesis of SCD is unclear, although there are a growing number of studies showing that the disease is characterized by increased iron and ROS concentrations, which can potentially contribute to the development of ferroptosis. Moreover, considering the complex pathogenesis of SCD, it would be reductive to say that the development of ferroptosis in SCD patients depends solely on these factors. In SCD subjects, excessive ROS production, dependent on different factors, leads to the accumulation of lipid peroxides, a condition that increases ferroptosis and results in worsening cellular damage [86]. SCD is related to iron overload resulting from both erythrocyte hemolysis, which causes accumulation of iron and heme in the blood, and continuous transfusions, which are one of the main strategies in treating the disease [87,88]. Excessive iron concentration contributes to lipid peroxidation and implements the activity of lipoxygenases and NOX1, enhancing ROS production through the Fenton reaction, promoting oxidative damage and SCD [89,90] (see Figure 2). A study by Menon et al. in SCD murine models showed that excess free heme stimulates HO-1 activity, causing increased iron (Fe^3+^) and lipid peroxidation, leading to ferroptosis and altered cardiac function. In detail from the study, it was found that the upregulation of HO-1 in SCD mouse models determined the increase in MDA (marker of lipid peroxidation) concentration and Ptgs2 (marker of ferroptosis) [90]. In addition, it was observed that GPx4 catalytic activity did not influence the process, a result in line with previous studies showing that iron-induced ferroptosis was not related to the inhibition of the enzyme [91]. The high iron concentration in SCD could upregulate the System Xc^−^, thus explaining the unchanged concentration of GSH and the GPx4 [92]. Menon et al. observed administration of human hemopexin prevents lipid peroxidation and ferroptosis in HbS mice, suggesting that heme reduction decreases cardiac ferroptosis in SCD [90]. A more recent study by Xi et al. shows the involvement of Nrf2 in the pathophysiology of SCD and the development of ferroptosis. In detail, Nrf2 depletion results in a reduction in proteins involved in the oxidative stress response, and a decrease in gene levels involved in heme and iron metabolism (SLC48A1 and HO-1 and Ftl1, FTH1, SLC40A1 and SLC11A2, respectively) and cysteine transport (SLC7A11); elevated levels of plasma heme and nonheme iron were found in the liver and spleen, and high serum levels of bilirubin (a product of heme degradation). In addition, Nrf2 depletion resulted in increased lipid peroxidation and the production of 4-hydroxynonenal [93]. In line with previous studies, the accumulation of iron and heme produces ROS through the Fenton reaction, increasing oxidative stress and promoting ferroptosis [67]. Recently, it has been reported that the transulfuration pathway (TSS) serves as an alternative pathway for cysteine production and glutathione synthesis, and this could reduce ROS and ferroptosis in SCD patients; in fact, given the excessive oxidative stress, SCD patients have low concentrations of cysteine required for GSH synthesis [94]. In this context, a study by Xi et al. highlighted the important contribution of the TSS pathway in regulating oxidative stress and ferroptosis in SCD erythroblasts. TSS pathway activation and physiological cystine supplementation protect SCD erythroblasts from ferroptosis by alleviating the disease. In detail, it was observed how dimethyl fumarate (DMF) treatment improves cystine uptake through stimulation of the transporter, contributes to the expression of factors involved in de novo cysteine synthesis (CBS), and stimulates GSH synthesis through pathways involving or not Nrf2 [95]. Therefore, iron chelation therapy and antioxidant molecules could improve the prognosis of individuals with SCD and reduce the progression of ferroptosis [96].

### 4.1. Ferroptosis and HbS Auto-Oxidation in SCD

As previously mentioned, the greater rate of auto-oxidation of HbS compared to HbA causes the production of free radicals, such as $O_{2}^{-}$, MetHb, or even ferril-hemoglobin, which result in increased oxidative stress to which erythrocyte cells are exposed [97]. The low oxygen concentrations to which sRBCs are exposed during their transit through blood vessels result in the polymerization of HbS, increase protein instability, and promote oxidative stress. Erythrocytes are cells equipped with important antioxidant systems, including superoxide dismutase (SOD), catalase (CAT), GPx4, etc. In detail, SOD is involved in the reaction that provides for the conversion of $O_{2}^{-}$, to H_2_O_2_ and oxygen. Increased peroxide concentration and iron overload in the erythrocyte environment directly promote the Fenton and Haber–Weiss reactions. This condition results in the onset of lipid peroxidation, damage to the erythrocyte membrane, and stimulates ferroptosis [98,99]. Lipid hydroperoxides, formed due to peroxidation, can interact with erythrocyte membrane proteins, such as band 3 protein. This interaction causes a functional alteration of the protein and its detachment from the membrane/cytoskeleton complex, resulting in the onset of erythrocyte membrane rupture and the production of microparticles [100,101]. Therefore, ferroptosis increases oxidative stress and cell death burden in SCD, worsening the clinical picture of the disease.

### 4.2. Ferroptosis, Vaso-Occlusion, and Hemolysis in SCD

Under conditions of low oxygen concentration, the polymerization of HbS and increased sRBC fragility led to increased cell hemolysis, releasing Hb, free heme, and iron into the blood [44]. In addition to intravascular hemolysis, the pathology is characterized by extravascular hemolysis at the level of several organs, such as the spleen and liver, which contribute to the production of ${NO}^{.}$ and ROS through Fenton and Haber–Weiss reactions. This results in the increased inflammatory state and accumulation of lipid peroxides, oxidative stress, and cell death [102,103,104]. More generally, in the blood, these conditions result in oxidative imbalance and the triggering of the ferroptotic pathway. NaveenKumar et al. showed that ferroptosis promotes platelet activation and subsequent aggregation, causing complications in blood clotting processes [105,106]. In this context, it should also be noted that platelet aggregation and activation are implicated in thrombus formation. As mentioned above, intra/extra-vascular hemolysis causes a reduction in NO concentration, a molecule implicated in the expression and activation of adhesion molecules on the cell surface. Specifically, NO inhibits the activities of fibrinogen and collagen receptors on the surface of platelets, inhibiting the aggregation process. NO depletion positively modulates the process of platelet aggregation, promoting thrombus formation [107]. Furthermore, Nader et al. demonstrated that microparticles released from sRBCs hemolysis may play an important role in macrovascular dysfunction in patients with SCD and that oxidative stress, induced by released radicals, would positively modulate eryptosis, inducing cell lysis and release of microparticles. This could lead to the activation of Toll-Like Receptor 4 (TLR4) and release of adhesion molecules and cytokines, which could contribute to worsening endothelial dysfunction [108]. Therefore, these factors can contribute to vaso-occlusion, kidney damage, and chronic hemolysis [109,110,111].

### 4.3. Ferroptosis and IRI in SCD

Ischemia–reperfusion injury (IRI) is a pathological condition triggered when the restoration of normal blood flow occurs in ischemic tissues [112,113,114]. IRI’s association with SCD has already been mentioned, and new studies show how ferroptosis will increase the cellular damage triggered by the pathology. In fact, the tissues affected by IRI, during SCD, are characterized by high concentrations of lipid peroxides, 4-hydroxynonenal (4-HNE), malondialdehyde (MDA), and iron, which contribute to the onset of ferroptosis, especially in organs such as kidneys, liver, brain, and lungs [113,115,116,117,118]. Several studies show the correlation between SCD, IRI, and ferroptosis; in addition, cases of IRI triggered by ferroptosis have been observed in anemic and nonanemic subjects, so inhibition of ferroptosis could potentially contribute to disease suppression [119,120]. A study by Menon et al. shows how excess circulating heme and decreased hemopexin, in a mouse model of sickle cell anemia, lead to increased free heme, which can enter cardiac cells by increasing heme oxygenase 1 (HMOX-1) expression. This results in increased levels of ferrous ions in cardiac cells, promoting the ferroptosis of cardiomyocytes [90]. This condition is further exacerbated by XO activity, which contributes to the release of free iron by promoting peroxidation, increasing iron-induced damage in cells, and contributing to the worsening of SCD [121]. High oxidative stress, iron accumulation at the mitochondrial level, and disruption of the BTB and CNC homology 1 (BACH1)/HMOX-1 axis contribute to ferroptosis. This may suggest how overexpression of System Xc^−^ in cardiac cells, due to increased GSH and involvement of GPx4, could inhibit feroptosis in SCD [122,123]. An enzyme of fundamental importance in cell protection is HMOX-1. This enzyme produces carbon monoxide and biliverdin, contributing to the anti-inflammatory and cytoprotective action [89,124]. Despite this, it has been seen that the overexpression of the enzyme can contribute to the accumulation of iron, ROS, lipid peroxidation, and ferroptosis [125,126]. Thus, HMOX-1 may have a dual effect on SCD, making new approaches and studies necessary to understand the role in SCD toxicology.

### 4.4. Ferroptosis and Inflammation in SCD

Inflammatory processes in SCD can have a dual action; these can indeed have a protective action or a harmful action and potentially depend on the infection or oxidative stress that characterizes the disease [127,128]. The inflammatory state in SCD can create a favorable environment for developing ferroptosis, worsening the disease state [89,105,129]. Ferroptosis and inflammation are interconnected processes that may favor each other and contribute to the progression of SCD. In addition, factors correlated with SCD, such as increased HbS, free iron and heme, vaso-occlusion, and cell lysis promote oxidative stress, vascular dysfunction, inflammation, and ferroptosis. The chronic inflammatory state that characterizes SCD is due to the activation of pro-inflammatory macrophages by free heme. Sharma et al. show that heme affects macrophage activities, such as immune cell recruitment ability and phagocytic capacity. In detail, the study shows how heme causes the reprogramming of macrophages through the activation of TLR4 and suppression of the transcription factor proliferator-activated receptor γ (PPARγ) and its peroxisome proliferator-activated receptor γ co-activator 1α (PGC1α). This causes defective recognition and uptake of apoptotic cells, mitochondrial remodeling, and reduced secretion of anti-inflammatory cytokines such as interleukin-4 (IL-4) and IL-10, inhibiting phagocytosis, resolution of inflammation, and tissue repair [130].

### 4.5. Ferroptosis and Transfusion in SCD

Transfusions are the first therapeutic strategy for patients with SCD, contributing on the one hand to increase blood oxygen transport capacity and on the other hand to reduce the phenomenon of vaso-occlusion. Despite this, different studies show that excessive transfusions contribute to the increase of iron and the development of ferroptosis in SCD subjects [56]. Blood is stored in the blood bank for a variable period, ranging from a few days to a few weeks, and during this period, RBCs can potentially undergo hemolysis [131,132]. In this context, reducing the speed of the phenomenon would help to make transfusions more effective as there would be more functioning erythrocytes, fewer adverse reactions and transfusion reactions, etc. Stolwijk et al. showed that the presence of GPx4 in RBCs is extremely variable and that about 75% of it is dependent on genetic traits. However, it should be noted that not all cells have the same enzymatic content; therefore, during the storage of blood in the blood bank, it is possible to witness the rupture of the cell membrane and erythrocyte hemolysis [133]. The alteration of erythrocyte metabolism during blood storage could implement cell dysfunction, causing oxygen saturation of Hb, leading to ROS production and triggering storage lesions [134]. A study conducted by D’Alessandro et al. on mouse models and human blood samples highlighted the role played by STEAP3 ferrireductase in the regulation of lipid peroxidation in stored blood samples, showing ferrireductase involvement in ferroptosis and how ferroptosis is related to the increase in Hb in transfusion recipients [135]. In addition, Youssef et al. show on mouse models that excessive transfusions contribute to the increase in erythrophagocytosis carried out by macrophages of the splenic red pulp (RPM), contributing to the implementation of ROS, lipid peroxidation, Ptgs2 gene expression and the reduction in RPM and therefore to the development of ferroptosis in cells [136]. In detail, the study showed that RPMs, a splenic population predominated in the murine models used in the study, strongly decreased following enhanced erythrophagocytosis due to the degradation of damaged RBCs. In addition, macrophages can potentially lead to an increase in the expression of HO-1 and ferritin, allowing cells to manage the increase in heme and iron resulting from erythrocyte hemolysis, thus protecting against oxidative stress. However, considering the enormous number of RBCs and the repeated increase in heme and iron, the increase in enzymes involved in the metabolism of iron and heme is insufficient [136,137].

## 5. Potential Therapeutic Strategies for Ferroptosis Involved in SCD

Given the connection between ferroptosis and SCD, research has been trying to understand how inhibition of the process could help achieve improvements in pathology. In SCD, the deficiency of antioxidant defenses and the accumulation of free iron make the environment susceptible to the development of ferroptosis. Among the potential therapeutic strategies, iron chelators, antioxidants, etc., could certainly help in improving the pathological status of the subjects.

Hydroxyurea (HU) is an approved drug for the treatment of SCD that acts through different mechanisms. HU could induce the production of HbF, inhibiting the polymerization of HbS and reducing the vaso-occlusive processes [138]. In addition, the drug has a myelosuppressive action, inducing a reduction in the number of activated blood cells and inhibiting inflammatory processes, and has the capacity to increase NO bioavailability by promoting vasodilation [139]. Furthermore, HU has antioxidant properties and shows iron-chelating ability, thus posing as a potent anti-ferroptotic agent by reducing ROS levels, increasing NO production, and activating the Nrf2 pathway in the endothelial cell line isolated from the vein of the umbilical cord (HUVEC) [140]. Recently, it has been shown that individuals with elevated serum ferritin levels or elevated hepatic iron concentrations can employ therapy based on deferiprone, deferoxamine, and deferasirox [141,142]. All these compounds have chelating properties and act by promoting metal elimination and reducing their toxicity, thereby inhibiting lipid peroxidation and ferroptosis [143].

Since SCD is characterized by the accumulation of ROS and RNS, antioxidants also possess an important role in the modulation and inhibition of HbS autoxidation reactions. In general, these molecules possess a broad spectrum of action and can act on different targets, including the modulation of GPx4, Nrf2 pathway, GSH production, etc. [144,145,146,147,148].

To conclude, two other important classes of molecules that may be of benefit in the treatment of SCD and ferroptosis are lipoxygenase inhibitors (ALOX) and inhibitors of the heme signaling pathway. We observe that individuals with SCD have high levels of ALOX5 in blood cells, favoring lipid peroxidation and inflammation [149,150]. Among the various compounds that may act in this field, zileuton acts by blocking leukotriene synthesis, inhibiting lipid peroxidation, ferroptosis, and adhesion of polymorphonucleates and sRBCs [151]; it also acts as an iron chelator. Baicalein can also potentially act in the treatment of SCD and can inhibit ferroptosis through the inhibition of ALOX [152].

ASP8731 is a novel inhibitor of BACH1, has anti-inflammatory, anti-vascular-occlusive activity, and induces the production of HbF [153,154]. This molecule could increase Nrf2 function and block the heme signaling pathway, inhibiting ferroptosis. In addition, this increments the synthesis of HMOX1 and FTH1 in liver cells, regulates GSH concentration in endothelial cells, and increases the expression of HMOX1 and γ-globin in mouse models [155,156]. 

## 6. Conclusions

To date, SCD remains a disease with a very complex pathogenesis that is dependent on different factors, among which the disease’s trigger is the mutation of the HBB gene that leads to the production of HbS. The mechanisms involved in SCD are different, and among them, HbS polymerization, vaso-occlusion, increased blood viscosity, endothelial dysfunction, ischemia–reperfusion injury, oxidative stress, and inflammation. It should be noted that the fragility of sRBCs causes an increase in the frequency of hemolysis, contributing to the increase in the concentration of iron, HbS, and free heme in the blood. These factors, together with the previous ones, strongly contribute to the production of ROS and the increase in oxidative stress. Recent studies have highlighted how this altered physiological condition contributes to ferroptosis, a new mechanism of iron-induced cell death characterized by lipid peroxidation, inhibition of GPx4, and the System Xc^−^.

In conclusion, further investigation of the molecular and cellular processes of SCD, especially the processes related to iron toxicity and oxidative stress, could further clarify the triggering of ferroptosis in the disease and allow the development of new therapeutic strategies, which could potentially include molecules with antioxidant activity, aimed at suppressing oxidative stress, iron accumulation and the progression of ferroptosis and disease.

## Figures and Tables

**Figure 1 ijms-26-03675-f001:**
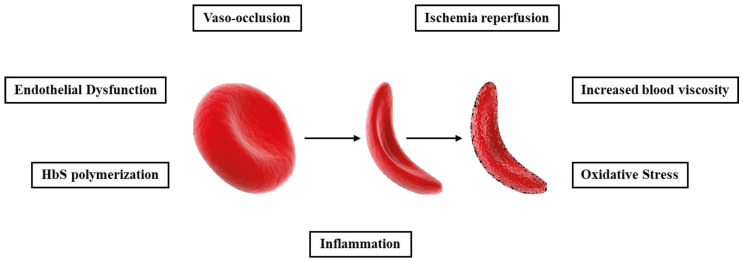
Pathophysiology of sickle cell disease. The image shows the gradual change in erythrocytes in SCD and the potential pathophysiological events that may be triggered due to the disease. Mutated HbS, under low oxygen pressure, tends to polymerize, increasing the fragility of sRBCs and promoting their lysis (normal RBC on the left and sickle cell on the right). This condition triggers an increase in HbS and iron in the bloodstream, contributing to the cascade of processes involved in the disease, such as oxidative stress, inflammation, endothelial dysfunction, etc.

**Figure 2 ijms-26-03675-f002:**
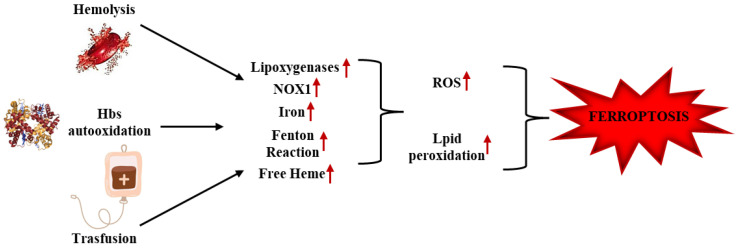
Implications of Ferroptosis in SCD. Hb auto-oxidation and erythrocyte hemolysis, two of the main features of SCD, and transfusions contribute to the accumulation of iron and heme in the blood. This condition, on the one hand, promotes the Fenton reaction leading to radical production, and on the other hand, promotes the activation of enzymes involved in radical production, such as lipoxygenase, NOX1, HO-1, etc. This leads to the accumulation of ROS, increased lipid peroxidation, and stimulation of ferroptosis.

## Data Availability

Not applicable.

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
