# Peer review of "Crosstalk Between Sickle Cell Disease and Ferroptosis"

_ijms, 2025, doi:10.3390/ijms26083675_

Round 1

Reviewer 1 Report

Comments and Suggestions for Authors

Description of sickle cell anemia requires greater care and accuracy. The aggregation or polymerization results in intracellular hemoglobin with increased viscosity as well as the abnormal membrane behavior and can give rise to abnormal morphology or sickle shape.  The authors imply that it is the abnormal morphology that gives rise to compromised blood flow. However, some red blood cells containing deoxygenated HbS polymer may not appear deformed but still exhibit abnormal blood flow. Section 3 introduces ferroptosis, iron metabolism and mitochondrial dysfunction. If RBCs undergo ferroptosis and do not contain mitochondria, what is the purpose of the discussion about mitochondria? The discussion should distinguish between ferroptosis relating to sRBC lysis and tissue ferroptosis giving rise to tissue specific pathology. The manuscript requires careful reading and use of references. The references are not always relevant to the discussion and errors in numbering make it very difficult to check the discussion with the citation. Figure 1 is not informative – the texts surrounding the images do not appear to relate to the drawings of the RBCs and the legend does not add further explanation.

     Line 12 - Abstract: HbS gives rise to a mutant hemoglobin that aggregates upon deoxygenation at physiologic conditions inside the red blood cell.  While the deoxygenated polymerized HbS does not bind oxygen cooperatively, unaggregated HbS or dilute HbS binds oxygen analogous to HbA.  It is not correct to say that HbS is a non-functional hemoglobin. 

     Line 28 - Introduction: sickle cell disease results from the production of a mutant hemoglobin rather than from abnormal production in hemoglobin.  Clarification is required.

     Line 37 – It is incorrect to say that the mutant HbS is a non-functioning hemogllboin.  Clarification is required.

     Line 39:  “deterioration of RBCs structure” requires clarification.  Do you mean the membrane defects resulting from repeated cycles of intracellular HbS polymerization and polymer melting as RBCs cycle between fully oxygenated state and low oxygenation as blood flows through the tissues?

     Line 42: The HbS polymer  consisting of long fibers in the red cell does not look like Hb crystals, unlike the intracellular crystals associated with HbC.  Please clarify.

     Line 62: Please clarify “the structural alteration of Hb” – Do you mean polymerization of deoxygenated HbS in contrast to no polymer formed in oxygenated HbS containing RBCs or HbA containing RBCs that lack structure?  Please clarify.

     Figure 1: The figure does not match the legend.  How the sequential change of RBCs from normal to sickle abnormal shape relates to a sequential behavior of HbS polymerization, endothelial dysfunction, vaso-occlusion, increased blood viscosity is suggested but unclear.  Are the words randomly placed or is there a specific order relating to the terms in the figure?  Is the dashed line around the abnormal RBC at the right signify an abnormal membrane? Please clarify.

     Line 94: The protective effect of HbF should be clarified. HbF composed of two alpha globin chains and two gamma globin chains (alpha2,gamma2) can form a mixed hybrid tetramer with HbS (alpha2,betaS2) to make (alpha2,betaS gamma).  At physiologic concentrations and upon deoxygenation, neither HbF (alpha2,gamma2) or the mixed hybrid (alpha2,betaS gamma) can enter the deoxygenated HbS polymer.  This results in the sparing effect of HbF and the resultant decrease in hemolgobin polymerization with increasing amounts of HbF which is more effective than HbA. “Slowing of the polymerization rate” requires further clarification  (Akinsheye, Alsultan et al., Blood 2011 118:19-27).

     Line 112:  “the altered structure of sRBCs” requires clarification. Does the altered structure refer to altered membrane, altered cell morphology or increased intracellular deoxygenated HbS polymer? Clarification required.

     Reference citations: Line 112 and references throughout. References do not always relate to the discussion where they are cited. Level of frustration is high trying to determine the supporting evidence for the review. For example, Ref. 21 on NO and SNO-Hb in RBCs does not relate to SCD or VOC.  Also, in line 183, Reiter is ref. 44 not Ref 45.

     2.4 Iron – references:

     Ref. 52: Not correct.  Paper on “Neuroinflammation in amyotrophic lateral sclerosis” does not relate to statement: Under normal conditions, the Hb released by RBCs binds, in the blood, haptoglobin (Hp) leading to the formation of a highly stable complex that hinders the release of iron and promotes its elimination through internalization in macrophages, thanks to the presence of the CD163 receptor [52].

     Ref. 44: Not correct. Paper on “Cell-free hemoglobin limits nitric oxide bioavailability in sickle-cell disease” does not mention macrophages as discussed: “the blood concentration of HbS increases, hindering the normal activity of macrophages, which on the one hand store iron to eliminate it, and on the other release it at the level of the blood flow; moreover, given the hydrophobicity of the molecule, it tends to accumulate at the level of cell membranes, implementing the production of ROS [44].”

     Ref. 55: not correct.  Paper on “Global burden of transfusion in sickle cell disease” does not discuss: “free heme, resulting from erythrocyte hemolysis, is able to react with O2, H2O2 and NO, resulting in the production of ROS and RNS and stimulating the Fenton reaction (Fe2+ + H2O2 → Fe3+ + ∙OH + OH‒) [55].”

     Ref. 57: Not correct. Here the authors found that the SCD PBMC transcriptome reflects high intracellular iron exposure, which is associated with inflammation and mortality but does not show: “studies in the literature show that about 30% of subjects with SCD may experience iron overload resulting from chronic transfusions [57]”.

     Ref. 58: Not correct. Study by Patel et al. examines iron status, iron deficiency and iron overload in sickle cell anemia patients and does not show: “how iron overload, linked to transfusions, causes serious diseases in different body districts, such as the heart, liver and pancreas, inducing cyto- toxicity on different cell lines [58].”

     Ref. 59:  Not correct. This review on ferroptosis does not mention sickle cell disease. Link to citation is not clear: “It is therefore essential to monitor ferritin levels in individuals with SCD to reduce their concentrations, through therapies that use iron chelators [59].”

    Reference 90 on line 331 should be reference 89 (Menon et al.)

    Reference 93 on line 344 should be reference 92 (Xi et al.)

    Ref. 95 on line 357 should be ref. 94.

     Line 228: Explain GPx4.  Define GPx4 and briefly state how the function of glutathione peroxidase 4 pathway relates to the RBC. 

     Line 229-231: RBC do not contain mitochondria. How does this activity relating to changes in mitochondria morphology and rupture of the outer mitochondrial membrane attributed to GPx4 relate to the RBC? Clarification required.

     Section 3.1. Iron metabolism and mitochondrial dysfunction:  How iron metabolism and mitochondrial dysfunction relates to RBCs that do not contain mitochondria requires further discussion. Also, please put in context the role of the RBC the concluding sentence: it has been observed how ferroptosis regulates the electron transport chain, inducing ROS production; in fact, the I-II- III complex of the mitochondrial respiratory chain (NADH-CoQ reductase complex, succinate dehydrogenase complex, cytochrome bc-1 complex, respectively) can produce ROS by damaging the mitochondrial membrane and triggering lipid peroxidation, inducing ferroptosis.

     Section 4. Ferroptosis and sickle cell disease: Please clarify the role of ferroptosis in hemolysis and the role of tissue specific ferroptosis that may result in tissue pathology. What are the mechanisms by which sRBCs lyse? Does ferroptosis account for all the increased lysis of sRBCs in sickle cell disease?  What other tissues are affected by ferroptosis in sickle cell disease? Does tissue specific ferroptosis contribute to increased pathology in sickle cell disease?

Author Response

Description of sickle cell anemia requires greater care and accuracy. The aggregation or polymerization results in intracellular hemoglobin with increased viscosity as well as the abnormal membrane behavior and can give rise to abnormal morphology or sickle shape.  The authors imply that it is the abnormal morphology that gives rise to compromised blood flow. However, some red blood cells containing deoxygenated HbS polymer may not appear deformed but still exhibit abnormal blood flow. Section 3 introduces ferroptosis, iron metabolism and mitochondrial dysfunction. If RBCs undergo ferroptosis and do not contain mitochondria, what is the purpose of the discussion about mitochondria? The discussion should distinguish between ferroptosis relating to sRBC lysis and tissue ferroptosis giving rise to tissue specific pathology. The manuscript requires careful reading and use of references. The references are not always relevant to the discussion and errors in numbering make it very difficult to check the discussion with the citation. Figure 1 is not informative – the texts surrounding the images do not appear to relate to the drawings of the RBCs and the legend does not add further explanation.

We thank the reviewer for the contribution. The work has been rechecked and fixed, image captions improved, and references fixed.

     Line 12 - Abstract: HbS gives rise to a mutant hemoglobin that aggregates upon deoxygenation at physiologic conditions inside the red blood cell.  While the deoxygenated polymerized HbS does not bind oxygen cooperatively, unaggregated HbS or dilute HbS binds oxygen analogous to HbA.  It is not correct to say that HbS is a non-functional hemoglobin. 

As suggested the abstract has been corrected.

     Line 28 - Introduction: sickle cell disease results from the production of a mutant hemoglobin rather than from abnormal production in hemoglobin.  Clarification is required.

As suggested the sentence “sickle cell disease results from the production of a mutant hemoglobin rather than from abnormal production in hemoglobin” has been corrected.

     Line 37 – It is incorrect to say that the mutant HbS is a non-functioning hemogllboin.  Clarification is required.

As suggested the sentence “mutant HbS is a non-functioning hemogllboin” has been corrected.

     Line 39:  “deterioration of RBCs structure” requires clarification.  Do you mean the membrane defects resulting from repeated cycles of intracellular HbS polymerization and polymer melting as RBCs cycle between fully oxygenated state and low oxygenation as blood flows through the tissues?

As suggested the sentence has been revised.

     Line 42: The HbS polymer  consisting of long fibers in the red cell does not look like Hb crystals, unlike the intracellular crystals associated with HbC.  Please clarify.

The sentence has been removed.

     Line 62: Please clarify “the structural alteration of Hb” – Do you mean polymerization of deoxygenated HbS in contrast to no polymer formed in oxygenated HbS containing RBCs or HbA containing RBCs that lack structure?  Please clarify.

The sentence has been corrected.

     Figure 1: The figure does not match the legend.  How the sequential change of RBCs from normal to sickle abnormal shape relates to a sequential behavior of HbS polymerization, endothelial dysfunction, vaso-occlusion, increased blood viscosity is suggested but unclear.  Are the words randomly placed or is there a specific order relating to the terms in the figure?  Is the dashed line around the abnormal RBC at the right signify an abnormal membrane? Please clarify.

The legend in Figure 1 has been revised.

     Line 94: The protective effect of HbF should be clarified. HbF composed of two alpha globin chains and two gamma globin chains (alpha2,gamma2) can form a mixed hybrid tetramer with HbS (alpha2,betaS2) to make (alpha2,betaS gamma).  At physiologic concentrations and upon deoxygenation, neither HbF (alpha2,gamma2) or the mixed hybrid (alpha2,betaS gamma) can enter the deoxygenated HbS polymer.  This results in the sparing effect of HbF and the resultant decrease in hemolgobin polymerization with increasing amounts of HbF which is more effective than HbA. “Slowing of the polymerization rate” requires further clarification  (Akinsheye, Alsultan et al., Blood 2011 118:19-27).

The period has been revised.

     Line 112:  “the altered structure of sRBCs” requires clarification. Does the altered structure refer to altered membrane, altered cell morphology or increased intracellular deoxygenated HbS polymer? Clarification required.

The sentence has been clarified

     Reference citations: Line 112 and references throughout. References do not always relate to the discussion where they are cited. Level of frustration is high trying to determine the supporting evidence for the review. For example, Ref. 21 on NO and SNO-Hb in RBCs does not relate to SCD or VOC.  Also, in line 183, Reiter is ref. 44 not Ref 45.

The references have been fixed, we apologize to the reviewer for the confusion.

     2.4 Iron – references:

     Ref. 52: Not correct.  Paper on “Neuroinflammation in amyotrophic lateral sclerosis” does not relate to statement: Under normal conditions, the Hb released by RBCs binds, in the blood, haptoglobin (Hp) leading to the formation of a highly stable complex that hinders the release of iron and promotes its elimination through internalization in macrophages, thanks to the presence of the CD163 receptor [52].

The reference has been fixed.

     Ref. 44: Not correct. Paper on “Cell-free hemoglobin limits nitric oxide bioavailability in sickle-cell disease” does not mention macrophages as discussed: “the blood concentration of HbS increases, hindering the normal activity of macrophages, which on the one hand store iron to eliminate it, and on the other release it at the level of the blood flow; moreover, given the hydrophobicity of the molecule, it tends to accumulate at the level of cell membranes, implementing the production of ROS [44].”

Reference 44 in line 218 has been deleted

     Ref. 55: not correct.  Paper on “Global burden of transfusion in sickle cell disease” does not discuss: “free heme, resulting from erythrocyte hemolysis, is able to react with O2, H2O2 and NO, resulting in the production of ROS and RNS and stimulating the Fenton reaction (Fe2+ + H2O2 → Fe3+ + ∙OH + OH‒) [55].”

The reference has been fixed.

     Ref. 57: Not correct. Here the authors found that the SCD PBMC transcriptome reflects high intracellular iron exposure, which is associated with inflammation and mortality but does not show: “studies in the literature show that about 30% of subjects with SCD may experience iron overload resulting from chronic transfusions [57]”.

The reference has been fixed and the text modified.

     Ref. 58: Not correct. Study by Patel et al. examines iron status, iron deficiency and iron overload in sickle cell anemia patients and does not show: “how iron overload, linked to transfusions, causes serious diseases in different body districts, such as the heart, liver and pancreas, inducing cyto- toxicity on different cell lines [58].”

The reference has been fixed.

     Ref. 59:  Not correct. This review on ferroptosis does not mention sickle cell disease. Link to citation is not clear: “It is therefore essential to monitor ferritin levels in individuals with SCD to reduce their concentrations, through therapies that use iron chelators [59].”

The reference has been fixed.

    Reference 90 on line 331 should be reference 89 (Menon et al.)

The reference has been fixed.

    Reference 93 on line 344 should be reference 92 (Xi et al.)

The reference has been fixed.

    Ref. 95 on line 357 should be ref. 94.

The reference has been fixed.

     Line 228: Explain GPx4.  Define GPx4 and briefly state how the function of glutathione peroxidase 4 pathway relates to the RBC. 

As suggested the period has been modified.

     Line 229-231: RBC do not contain mitochondria. How does this activity relating to changes in mitochondria morphology and rupture of the outer mitochondrial membrane attributed to GPx4 relate to the RBC? Clarification required.

As suggested the period has been improved and clarified.

     Section 3.1. Iron metabolism and mitochondrial dysfunction:  How iron metabolism and mitochondrial dysfunction relates to RBCs that do not contain mitochondria requires further discussion. Also, please put in context the role of the RBC the concluding sentence: it has been observed how ferroptosis regulates the electron transport chain, inducing ROS production; in fact, the I-II- III complex of the mitochondrial respiratory chain (NADH-CoQ reductase complex, succinate dehydrogenase complex, cytochrome bc-1 complex, respectively) can produce ROS by damaging the mitochondrial membrane and triggering lipid peroxidation, inducing ferroptosis.

In Section 3 the discussion of mitochondria and in general the characteristics of ferroptosis are not related exclusively to red blood cells. The section serves to describe the general characteristics of the process and to understand how it manifests itself in cells. In addition, ferroptosis is also discussed in non-erythrocytic cells in the text, so the explanation of the process may help the reader in understanding the work.

    Section 4. Ferroptosis and sickle cell disease: Please clarify the role of ferroptosis in hemolysis and the role of tissue specific ferroptosis that may result in tissue pathology. What are the mechanisms by which sRBCs lyse? Does ferroptosis account for all the increased lysis of sRBCs in sickle cell disease?  What other tissues are affected by ferroptosis in sickle cell disease? Does tissue specific ferroptosis contribute to increased pathology in sickle cell disease?

Section 4 has been implemented.

Reviewer 2 Report

Comments and Suggestions for Authors

An interesting paper talking about Sickle Cell Anaemia (SCA) and ferroptosis. The review is well organized and written, the authors start by describing SCA physiopathology characteristics, then they concentrate on the ferroptosis general mechanisms to end with the relationship between SCA and ferroptosis.

The “weight” and space given to the 3 chapters is equivalent, I suggest that the authors be a little less prolific on the generalities of SCA and ferroptosis to give more space to ferroptosis in the SCD.

It will be interesting discuss more in details the effects of excess heme versus free iron versus HbS autoxidation in SCD ferroptosis: do they active the same pathway or different pathways? Do they converge finally in the same one? A recent paper (Sharma R et al Blood 2023) shows that heme drives a coordinated functional and metabolic reprogramming of macrophages.

What about ferroptosis in the SCA concerning the different organs? The impact in the circulation? A relation with vaso-occlusion and/or haemolysis? Liver? Spleen? Several studies have shown that ferroptosis contributes to IRI in various organs.

Could the authors develop in more detail the relationship between ferroptosis and inflammation in SCA? How ferroptosis contributes to inflammation in SCA? How it Influence the response of immune cells (leukocytes, macrophages, neutrophils…)?

It will be interesting discuss deeply if and how ferroptosis offer novel therapeutic targets in SCA and how actual drug therapies (ex HU, iron chelators…) play a role in the modulation of ferroptosis. Several inhibitors of ferroptosis are described in other pathologies in particular in cancer treatment, do they offer potential therapeutic benefits for SCA?

Pay attention: there are some imprecisions, as in “sickling of erythrocytes results in the exposure of… basal cell adhesion molecule-1/Lutheran (B-CAM-1/Lu), … and intercellular adhesion molecule-4 (ICAM-4), adhesion molecules that are not normally expressed in the extracellular region of the erythrocyte membrane [7].” Note that Lu/BCAM and ICAM4 are express on the surface of the normal RBC, they are carriers of respectively of Lutheran and Landsteiner-Wiener blood group systems! In SCA Lu/BCAM is constitutively activated on a large proportion of the erythrocytes through phosphorylation resulting in an abnormal RBC adhesion.

Author Response

An interesting paper talking about Sickle Cell Anaemia (SCA) and ferroptosis. The review is well organized and written, the authors start by describing SCA physiopathology characteristics, then they concentrate on the ferroptosis general mechanisms to end with the relationship between SCA and ferroptosis.

We thank the reviewer for input and suggestions.

The “weight” and space given to the 3 chapters is equivalent, I suggest that the authors be a little less prolific on the generalities of SCA and ferroptosis to give more space to ferroptosis in the SCD.

As suggested the paragraph on the correlation between SCD and Ferroptosis has been expanded.

It will be interesting discuss more in details the effects of excess heme versus free iron versus HbS autoxidation in SCD ferroptosis: do they active the same pathway or different pathways? Do they converge finally in the same one? A recent paper (Sharma R et al Blood 2023) shows that heme drives a coordinated functional and metabolic reprogramming of macrophages.

As suggested the effects of excess heme, free iron and HbS autoxidation have been explained.

What about ferroptosis in the SCA concerning the different organs? The impact in the circulation? A relation with vaso-occlusion and/or haemolysis? Liver? Spleen? Several studies have shown that ferroptosis contributes to IRI in various organs.

As suggested the discussion has been expanded through the addition of additional paragraphs.

Could the authors develop in more detail the relationship between ferroptosis and inflammation in SCA? How ferroptosis contributes to inflammation in SCA? How it Influence the response of immune cells (leukocytes, macrophages, neutrophils…)?

As suggested the discussion has been expanded through the addition of additional paragraph.

It will be interesting discuss deeply if and how ferroptosis offer novel therapeutic targets in SCA and how actual drug therapies (ex HU, iron chelators…) play a role in the modulation of ferroptosis. Several inhibitors of ferroptosis are described in other pathologies in particular in cancer treatment, do they offer potential therapeutic benefits for SCA?

As suggested the discussion has been expanded.

Pay attention: there are some imprecisions, as in “sickling of erythrocytes results in the exposure of… basal cell adhesion molecule-1/Lutheran (B-CAM-1/Lu), … and intercellular adhesion molecule-4 (ICAM-4), adhesion molecules that are not normally expressed in the extracellular region of the erythrocyte membrane [7].” Note that Lu/BCAM and ICAM4 are express on the surface of the normal RBC, they are carriers of respectively of Lutheran and Landsteiner-Wiener blood group systems! In SCA Lu/BCAM is constitutively activated on a large proportion of the erythrocytes through phosphorylation resulting in an abnormal RBC adhesion.

The sentence was changed and the reference modified.

Round 2

Reviewer 1 Report

Comments and Suggestions for Authors

Authors have addressed some of the concerns raised in the original review and extended the text to provide further clarification.

Concerns:

Line 44: “alter cell structure by implementing its fragility” requires clarification. Do you mean alter cell rheology or compromise blood flow or cause obstruction to blood flow or increase cell fragility?

Figure 1 legend, line 82: “implementing the fragility” requires clarification. DO you mean “tends to polymerize increasing the fragility of sRBCs….”

Line 101: “slowing down the polymerization process” requires clarification. Do you mean decreasing the kinetics and extent of polymerization or decreasing the polymerization tendency?

Line 102: “HbF cannot interact with HbS in the deoxygenated state, preventing its polymerization” requires clarification. Do you mean HbF cannot participate in the formation of the deoxygenated HbS polymer state? What does “its polymerization” mean?

Much of new Sections 4.1 to 4.4 resemble the discussion in narrative content and in the selection and order of references cited from Fortuna et al., Current Research in Toxicology 2024. Please provide further clarification. The authors should carefully read the references cited and use their own narrative to provide a more accurate description and/or use alternative references. 

For example, Lines 390-392 – Reference 97 describes increased oxidative stress in HbS containing RBCs, MP formation and vascular toxicity in the Townes-SS mice.  The authors should clarify how do these activities relate to RBCs from sickle cell disease patients

Lines 393-400: Contrary to the description of HbS behavior in erythrocytes, Ref. 98 and Ref 99 do not focus on intact RBCs. Ref 98 examines cell free hemoglobin solutions and cellular toxicity in mouse lung epithelial cells. Ref. 99 is a review of myoglobin and hemoglobin and discusses hemoglobin based oxygen carriers.

Line 415: The authors list six references  [42,89,100,102-104]. Which of these will be the most useful to provide support and more details that would be useful to the reader?

42, 89,100, 102, 103 are review articles. Are these all of equal importance and necessary?

Line 417-418: NaveenKumar et al. showed that ferroptosis promotes platelet activation and subsequent aggregation, causing complications in blood clotting processes [105-107]. Ref 107 is not authored by NaveenKumar.

Line 531: “that can be beneficially used” implies that these treatments are already in use.  Do you mean “that may be of benefit in treatment of SCD” or “that have been proposed to be of benefit in treatment of SCD”? – Clarification required.

Author Response

Authors have addressed some of the concerns raised in the original review and extended the text to provide further clarification.

We thank the reviewer for comments and suggestions, that allowed us to improve the review.

Concerns:

Line 44: “alter cell structure by implementing its fragility” requires clarification. Do you mean alter cell rheology or compromise blood flow or cause obstruction to blood flow or increase cell fragility?

Line 44, the sentence has been revised as follows: The transit of erythrocytes within blood vessels causes cells to be exposed to oxy-gen-deficient tissue regions, resulting in the polymerization of HbS that alter erythrocyte structure (morphology) and increase cellular fragility”

Figure 1 legend, line 82: “implementing the fragility” requires clarification. DO you mean “tends to polymerize increasing the fragility of sRBCs….”

Figure 1 legend, line 82, as suggested “implementing” has been replaced by “increasing”.

Line 101: “slowing down the polymerization process” requires clarification. Do you mean decreasing the kinetics and extent of polymerization or decreasing the polymerization tendency?

As suggested line 101 has been clarified: “High levels of HbF are implicated in reducing the concentration of HbS, slowing down the rate and tendency of polymerization”.

Line 102: “HbF cannot interact with HbS in the deoxygenated state, preventing its polymerization” requires clarification. Do you mean HbF cannot participate in the formation of the deoxygenated HbS polymer state? What does “its polymerization” mean?

As suggested line 102 has been revised and clarified: “HbF cannot participate in the formation of the deoxygenated polymeric state of HbS”.

Much of new Sections 4.1 to 4.4 resemble the discussion in narrative content and in the selection and order of references cited from Fortuna et al., Current Research in Toxicology 2024. Please provide further clarification. The authors should carefully read the references cited and use their own narrative to provide a more accurate description and/or use alternative references. 

We thank the reviewer for the suggestions. Indicated section have been improved and alternative references have been added.

For example, Lines 390-392 – Reference 97 describes increased oxidative stress in HbS containing RBCs, MP formation and vascular toxicity in the Townes-SS mice.  The authors should clarify how do these activities relate to RBCs from sickle cell disease patients

As suggested the reference 97 has been replaced. 

Lines 393-400: Contrary to the description of HbS behavior in erythrocytes, Ref. 98 and Ref 99 do not focus on intact RBCs. Ref 98 examines cell free hemoglobin solutions and cellular toxicity in mouse lung epithelial cells. Ref. 99 is a review of myoglobin and hemoglobin and discusses hemoglobin based oxygen carriers.

As suggested the references 98,99 have been replaced. 

Line 415: The authors list six references  [42,89,100,102-104]. Which of these will be the most useful to provide support and more details that would be useful to the reader?

42, 89,100, 102, 103 are review articles. Are these all of equal importance and necessary?

As suggested references 42,89,100 have been deleted.

Line 417-418: NaveenKumar et al. showed that ferroptosis promotes platelet activation and subsequent aggregation, causing complications in blood clotting processes [105-107]. Ref 107 is not authored by NaveenKumar.

As suggested the references to the line 417-418 has been fixed and the discussion expande: “In this context, it should also be noted that platelet aggregation and activation are implicated in thrombus formation. As mentioned above, intra/extra-vascular hemolysis causes the reduction of NO concentration, a molecule implicated in the expression and activation of adhesion molecules on the cell surface. Specifically, NO inhibits the activities of fibrinogen and collagen receptors on the surface of platelets, inhibiting the aggregation process. NO depletion positively modulates the process of platelet aggregation, promoting thrombus formation [107]”.

Line 531: “that can be beneficially used” implies that these treatments are already in use.  Do you mean “that may be of benefit in treatment of SCD” or “that have been proposed to be of benefit in treatment of SCD”? – Clarification required.

Line 531, as suggested has been revised and clarified: “that may be of benefit in the treatment of SCD”

Round 3

Reviewer 1 Report

Comments and Suggestions for Authors

The authors address most of the concerns raised in the previous review.